# Soluble ITGaM and ITGb2 Integrin Subunits Are Involved in Long-Term Pulmonary Complications after COVID-19 Infection

**DOI:** 10.3390/jcm12010342

**Published:** 2023-01-01

**Authors:** Kamil Siekacz, Anna Kumor-Kisielewska, Joanna Miłkowska-Dymanowska, Małgorzata Pietrusińska, Krystian Bartczak, Sebastian Majewski, Adam Stańczyk, Wojciech J. Piotrowski, Adam J. Białas

**Affiliations:** 1Department of Pneumology, Medical University of Lodz, 90-419 Lodz, Poland; 2Department of Clinical Pharmacology, Medical University of Lodz, 90-419 Lodz, Poland; 3Department of Pulmonary Rehabilitation, Regional Medical Center for Lung Diseases and Rehabilitation, 91-520 Lodz, Poland

**Keywords:** ITGaM, CD11b, ITGb2, CD18, integrins, biomarkers, COVID-19, long COVID-19, post-COVID-19, pulmonary complications, fibrosis

## Abstract

(1) Introduction: The role of soluble integrins in post-COVID-19 complications is unclear, especially in long-term pulmonary lesions. The purpose of this study was to investigate the association between soluble ITGa2, ITGaM and ITGb2 integrin subunits and long COVID-19 pulmonary complications. (2) Methodology: Post-COVID-19 patients were enrolled. According to the evidence of persistent interstitial lung lesions on CT, patients were divided into a long-term pulmonary complications group (P(+)) and a control group without long-term pulmonary complications (P(−)). We randomly selected 80 patients for further investigation (40 subjects for each group). Levels of ITGa2, ITGaM and ITGb2 integrin subunits were determined by ELISA assay. (3) Results: The serum concentration of sITGaM and sITGb2 were significantly higher in the P(+) group (sITGaM 18.63 ng/mL [IQR 14.17–28.83] vs. 14.75 ng/mL [IQR 10.91–20] *p* = 0.01 and sITGb2 10.55 ng/mL [IQR 6.53–15.83] vs. 6.34 ng/mL [IQR 4.98-9.68] *p* = 0.002). We observed a statistically significant correlation between sITGaM and sITGb2 elevation in the P(+) group (R = 0.42; *p* = 0.01). Patients from the P(+) group had a lower (1.82 +/−0.84 G/L) lymphocyte level than the P(−)group (2.28 +/−0.79 G/L), *p* = 0.03. Furthermore, we observed an inverse correlation in the P(−) group between blood lymphocyte count and sITGb2 integrin subunit levels (R = −0.49 *p* = 0.01). (4) Conclusions: Elevated concentrations of sITGaM and sITGb2 were associated with long-term pulmonary complications in post-COVID-19 patients. Both sITGaM and sITGb2 may be promising biomarkers for predicting pulmonary complications and could be a potential target for therapeutic intervention in post-COVID-19 patients.

## 1. Introduction

Integrins are pivotal receptors in leukocyte adhesion [1]. We have focused on the role of leukocyte-specific β2 integrins, mainly expressed and exposed on the white cell surface, during the COVID-19 disease. This group of transmembrane receptors is composed of a variable α (from CD11a to CD11d) and a constant β (CD18; ITGb2) subunit. The expression of subunit α regulates the surface number of leukocyte-specific β2 integrins, as subunit β is continuously expressed in leukocyte cells [2,3]. Leukocyte-specific β2 integrins are presented in an inactive conformational form with low affinity. Conformational activation resembles a switchblade-like motion in which integrin extends and the binding packet “opens” for ligands [4,5]. Integrins mediate the inflammatory response and contribute to anchoring leukocytes in endothelium, which allows their diapedesis. They play critical roles in leukocyte chemotaxis during cell adhesion and migration [6]. Moreover, β2 integrins are inherent in (1) phagocytosis of opsonized pathogens and immune complexes; (2) modulation of other receptors, for example toll-like receptor (TLR); (3) interaction between immune cells, and immune cells and target cells; (4) differentiation of white cells via stimulation of transcription factor function [2].

Integrins are directly involved in severe acute respiratory syndrome coronavirus 2 (SARS-CoV-2) infection via the arginylglycylaspartic acid motif (RGD motif) [7]. The RGD motif is similar to the angiotensin-converting enzyme 2 (ACE2) receptor. SARS-CoV-2’s spike protein (S protein) utilizes the ACE2 receptor to enter epithelial-like cells. Therefore, the integrin RGD motif mediates ACE2-independent cell entry of the SARS-CoV-2 virus [8]. On the other hand, integrins could suppress virus entry by shielding the interaction between S protein and ACE2 [9]. Integrin-mediated cell access is possible in their active conformation [10]. Viral binding via integrins contributes to the dysregulation of integrin pathways with consequent cell damage [11]. Importantly, integrins are not simple adhesion molecules. They are associated with downstream signaling cascades, modulate gene-transcription programs, and facilitate phagocytosis and extracellular matrix reorganization [12]. 

Recently, soluble serum forms of β2 integrins have gained significance. The expression of a soluble form of integrin Mac-1(sITGaM/sITGb2; sCD11b/sCD18) reveals a new direction in integrin research [13]. The functional form of the sITGaM/sITGb2 extracellular integrins domain arises during the detachment of leukocytes via enzymatic cleavage [14]. An additional source of sITGaM/sITGb2 is the membrane release stimulated by tumor necrosis factor alpha (TNF-alpha) [15]. Altered levels of soluble integrin beta 2 subunit (sITGb2, sCD18,) are associated with spondylarthritis activity and sepsis outcomes. A decreased concentration of sITGb2 correlates with alcoholic hepatitis and spondylarthritis. Moreover, sITGb2 has been established as a biomarker of fatal sepsis outcome [16,17,18]. 

The emerging problem seems to be long-term health complications after COVID-19 recovery [19]. According to the European Respiratory Society Statement, long COVID-19 is defined as “signs and symptoms that continue or develop after acute COVID-19 and post-COVID-19 syndrome, encapsulates those with symptoms persisting > 12 weeks” [20]. Common symptoms of long COVID-19 involve fatigue (58%), headache (44%), attention disorder (27%), hair loss (25%), and dyspnoea (24%). Meta-analysis of the long-term complications indicates that abnormal chest X-ray and/or CT results are revealed in 34% (27–42%) of post-COVID-19 patients after more than two weeks of observation [21,22].

This study aimed to investigate the association between main soluble integrin subunits and long COVID-19 pulmonary complications. Our research aimed to evaluate the significance of sITGa2, sITGaM and sITGb2 integrins as biomarkers and their involvement in immune response after COVID-19.

## 2. Materials and Methods

### 2.1. Subjects

We enrolled 283 patients (mean age = 55 ± 12) from the Outpatient Clinic and Department of Pneumology of the Medical University of Lodz from 2020 to the end of 2021, who were recovered from COVID-19. In all patients, viral infection was confirmed by real-time polymerase chain reaction test (RT-PCR). From the entire study cohort, we selected 80 patients for further investigation (40 subjects for each group), as described below. Pulmonary manifestation of long COVID-19 was defined by lung lesions, with or without decreased parameters of pulmonary function tests (PFTs), which persisted after approximately 3 months after recovery from active COVID-19. This group was labelled as P(+). The control group consisted of patients who had recovered from COVID-19 and neither presented lung lesions nor decrease in PFTs in a 3-month follow-up. Patients in this group were selected randomly using the web-based tool served by the research randomizer page (https://www.randomizer.org, accessed on 23 March 2022). This group was labelled as P(−). Smokers and ex-smokers were defined according to recommendations of the Center for Disease Control and Prevention [23]. We performed a comprehensive assessment of the participants, presented in Table 1. The Experiment design was shown in Figure 1.

### 2.2. Pulmonary Function Tests

Spirometry and the single-breath transfer factor of the lung for carbon monoxide (TLCO) measurements were performed using a Lungtest 1000 system (MES, Cracow, Poland) according to ATS/ERS standards [24]. Forced expiratory volume in 1 s (FEV1), forced vital capacity (FVC), FEV1/FVC% and TLCO corrected for hemoglobin concentration were recorded.

### 2.3. Samples

Venous blood samples were collected by venipuncture into tubes with K2EDTA and tubes for preparing serum with gel (total volume 4.5 + 5 mL). Samples for serum were allowed to clot and after 30 min were centrifuged at approximately 1000× *g*, at 4 °C for 10 min (Centrifuge MPW 223e). The serum was then transferred to an Eppendorf tube and stored at −80 °C for later use. Morphology and biochemistry measurements were performed using a Sysmex 2000XN and a Beckman Coulter au480, respectively.

### 2.4. ELISA

Serum levels of sITGa2, sITGaM and sITGb2 were measured by an Enzyme-linked Immunosorbent Assay Kit (Cloud-Clone Corp., Houston, TX, USA). The manufacturer’s protocol was followed. Detection was performed on the microplate reader (Microplate Reader BioTek 800 Elx) and measurement was conducted at 450 nm immediately. The minimum detection range of sITGaM and sITGb2 could be less than 13.3 pg/mL (0.0133 ng/mL) and 116 pg/mL (0.116 ng/mL), respectively. 

### 2.5. Statistical Analysis

Statistical analysis was performed using Statistica v13.3 2017 TIBCO software for Windows OS. Continuous data were presented as the mean with standard deviation (SD) or median with interquartile range (IQR), depending on the distribution of data. Variables were compared using the unpaired Student’s *t*-test and the U Mann–Whitney test with continuity correction, depending on data normality and homogeneity of variance. Pearson’s correlation test was used for correlation analyses.

## 3. Results

### 3.1. Pulmonary Function Test Results

For technical reasons, in the P(+) group the spirometry results were was obtained from 36 patients and in the P(−) group in 37 patients. The results are presented in Table 2. We noted a statistically significant difference in FVC, FEV1 and TLCO between the investigated groups. 

### 3.2. Analysis of Integrin Subunits Profiles

Serum concentrations of sITGaM, sITGa1 and sITGb2 were analyzed. The sITGa1 concentration was under the detection range in both groups of patients (data not shown). 

The serum concentrations of sITGaM and sITGb2 were significantly higher in patients with long-term pulmonary complications (P(+)) as compared with the control P(−) group (sITGaM 18.63 ng/mL IQR 14.17–28.83 vs. 14.75 ng/mL IQR 10.91–20 *p* = 0.01 and sITGb2 10.55 ng/mL IGR 6.53–15.83 vs. 6.34 ng/mL IGR 4.98-9.68 *p* = 0.002) (Figure 1a,b).

Moreover, we observed a statistically significant correlation between sITGaM and sITGb2 elevation in the P(+) group. Pearson’s correlation factor was R = 0.42, *p* = 0.01 (Figure 2). In the P(−) group, this correlation was not observed (data not shown).

Additionally, we noted a statistically significant difference in lymphocyte blood count levels between the studied groups (Figure 3). Patients in the P(+) group had lower (mean = 1.82+/−0.84 G/L) lymphocyte levels than the P(−) group (mean = 2.28+/−0.79 G/L). Furthermore, we observed an inverse correlation between peripheral blood lymphocyte levels and sITGb2 integrin subunit levels in the P(−) group. Pearson’s correlation factor in the P(−) group was R = −0.49, *p* = 0.01 (Figure 4). In the P(+) group, this correlation was not observed (data not shown).

## 4. Discussion

Levels of sITGaM and sITGb2 in SARS-CoV2 infection were investigated here for the first time. In reviewing the literature, no data were found on the role of integrin soluble subunits in COVID-19. Thus, their role in long COVID-19 pulmonary complications is unexplained.

This study suggests that sITGaM and sITGb2 (sCD11b and sCD18) are associated with long-term pulmonary complications in post-COVID-19 patients. A possible explanation for these results might be that integrins are secreted by neutrophils to the environment in the presence of pathogens [25]. Additionally, integrin subunits are cleavaged by sororicides during leucocyte detachment [14]. ITGaM (CD11b) is necessary for the development of inflammation during pulmonary infections [26]. Moreover, sCD11a/b/c and sCD18 integrin subunits could dimerize and/or orchestrate in complexes [15]. They are fully functional and could bind to their ligands and microorganisms [13]. The findings mentioned above might explain the correlation between sITGaM and sITGb2 noted in our study. We observed elevated sITGaM and sITGb2 integrin subunits in the long-term pulmonary complications group. These data are consistent with Kragstrup et al. who demonstrated that sITGb2 (sCD18) levels are associated with leukocyte migration and the release of cytokines. This study supports evidence that intensive monocyte migration increases the serum level of sITGb2. Simultaneously, macrophages infiltrate tissues, causing damage. Therefore, sITGb2 is a serum biomarker of monocyte migration. In this context, sITGb2 is a crucial regulator of chemotaxis and migration to infected tissue [17]. 

Another important finding of our study was the inverse correlation between peripheral blood lymphocyte levels and sITGb2 serum levels. Through the feedback loop, a high concentration of sITGb2 inhibits further lymphocyte tissue infiltration by binding and blocking endothelial migration receptors. Soluble integrin subunits bind to their ligands and competitively inhibit integrin receptors on the target cells and immune cell surfaces [27]. In this way, blocked receptors cannot function properly, including in their differentiation, migration, stimulation, and activation of T-lymphocytes (TL) [2]. Specifically, the subset of CD8+ T cells that include both the active virus-specific cytotoxic TL (CTL) and the virus-specific memory CTL populations are inactivated. Therefore, the cell-killing ability is diminished and, in consequence, the viral infection continues to progress [28]. On the other hand, CD11b−/− polymorphonuclear leukocytes are less prone to apoptosis and CD11b−/− mice are more susceptible to developing autoimmune diseases [29]. The CD18 knockout impairs wound healing [30]. 

Likewise, the results of our study suggest that increased levels of sITGaM (sCD11b) and sITGb2 (sCD18) are associated with long-term pulmonary complications after recovering from COVID-19. We hypothesize that the rapid shedding of a large amount of sITGaM and sITGb2 integrin subunits from the monocyte surface aggravates lung tissue inflammation. Excessively released integrin sITGb2 subunits impair lymphocytes’ regulatory functions. A decreased count of lymphocytes leads to prolonged focal inflammation and, consequently, lung tissue remodeling. Moreover, crosslinking of integrin subunits induces the activation of granulocytes. Increased levels of over-reactive neutrophils release enzymes and cytokines [31]. Exocytosis of azurophilic granules and degranulation may enhance micro-vessel injury and focal tissue damage [32] (Figure 5).

On the other hand, fluctuations of sITGb2 levels are sufficient to affect the infection state. Inadequate sITGb2 shedding and/or cellular expression processes could cause chronic inflammation. Equally, decreased or increased levels of sITGb2 could be involved in pulmonary lesions. The outcome depends on several variables including monocyte dysfunction, metalloproteinase activity, detachment enzyme activity, ITGaM/ITGb2 levels, and expression of their ligands [16]. Therefore, further research on these processes is needed.

These findings might have important implications for screening, preventing, and treating long-term pulmonary complications [9,10,33]. Elevated levels of integrins could be utilized as biomarkers of interstitial lung lesions [33]. Levels of integrins could be used to predict the course of the disease and its outcomes. Early identification of patients susceptible to lung lesions could improve their further health care. Integrins should be considered a target for fibrosis inhibitors [34], especially in pulmonary fibrosis [35]. Moreover, antifibrotic agents could be used against integrin ligands to decrease the activation of fibrosis cascades [36]. Integrins have a crucial role in leucocyte activation and migration [6]. Our study suggests that disturbances in soluble integrin subunits homeostasis may contribute to immunological dysregulation [37]. This probably has an association with T lymphocyte immune response [2]. Investigations of serum integrin subunits play an important role in elucidating the pathophysiology of interstitial lung lesions after COVID-19 [9]. 

Despite the novel findings of the present study, it has several limitations. This study was limited by the absence of a healthy control group and the small size of the investigated groups. Due to technical reasons, some clinical and laboratory data were not available or were limited. The generalizability of these results is subject to specific limitations: wide follow-up range, heterogeneity of patients, and pulmonary complications. Subtypes of lymphocytes were not identified. These factors might have introduced a selection bias for some data. 

## 5. Conclusions

Our study’s findings suggest an association between sITGaM, sITGb2, and long-term pulmonary complications in post-COVID-19 patients. The correlated elevation of soluble integrin subunits in the P(+) group compared to the P(−) control group implies an assumption that they may be promising biomarkers for predicting pulmonary complications and could be a potential therapeutic target in post-COVID-19 patients. In addition, our study results support the hypothesis that elevated sITGb2 integrin subunit levels inhibit T-lymphocyte-dependent immune response. Additional research is warranted to validate the present study findings. 

## Data Availability

Not applicable.

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
