# Peer review of "Soluble ITGaM and ITGb2 Integrin Subunits Are Involved in Long-Term Pulmonary Complications after COVID-19 Infection"

_jcm, 2023, doi:10.3390/jcm12010342_

Round 1
Reviewer 1 Report
Interesting and novel paper studying the relationship of ITGaM and ITGb2 integrins for long covid syndrome.
Major corrections:
Study methodology is not clear. Pulmonary manifestation is defined by imaging - is it at diagnosis or follow up? Was imaging repeated after 3 months follow up as well? Did all the patients at 3 months follow up had symptoms of long covid?
What is the reason to decrease sample size to 80? How were they further selected/filtered?
Minor changes:
Commas are mentioned in most of the places instead of decimal point which needs to be replaced

Author Response
Dear Editor,
We are very grateful for all the valuable Reviewers’ comments. We hope that the changes we made in our manuscript improved it satisfactorily. We presented our responses in attachament.

Reviewer 2 Report
I read the manuscript with interest.
The authors should see the attached file and the highlighted text and comment and change accordingly.

Author Response

(The authors gave the same response as above.)
